# Structures of tweety homolog proteins TTYH2 and TTYH3 reveal a Ca$^{2+}$-dependent switch from intra- to intermembrane dimerization

Baobin Li ⓘ [1,2,3,4], Christopher M. Hoel ⓘ [1,2,3,4] & Stephen G. Brohawn ⓘ [1,2,3✉]

Tweety homologs (TTYHs) comprise a conserved family of transmembrane proteins found in eukaryotes with three members (TTYH1-3) in vertebrates. They are widely expressed in mammals including at high levels in the nervous system and have been implicated in cancers and other diseases including epilepsy, chronic pain, and viral infections. TTYHs have been reported to form Ca$^{2+}$- and cell volume-regulated anion channels structurally distinct from any characterized protein family with potential roles in cell adhesion, migration, and developmental signaling. To provide insight into TTYH family structure and function, we determined cryo-EM structures of *Mus musculus* TTYH2 and TTYH3 in lipid nanodiscs. TTYH2 and TTYH3 adopt a previously unobserved fold which includes an extended extracellular domain with a partially solvent exposed pocket that may be an interaction site for hydrophobic molecules. In the presence of Ca$^{2+}$, TTYH2 and TTYH3 form homomeric cis-dimers bridged by extracellularly coordinated Ca$^{2+}$. Strikingly, in the absence of Ca$^{2+}$, TTYH2 forms trans-dimers that span opposing membranes across a ~130 Å intermembrane space as well as a monomeric state. All TTYH structures lack ion conducting pathways and we do not observe TTYH2-dependent channel activity in cells. We conclude TTYHs are not pore forming subunits of anion channels and their function may involve Ca$^{2+}$-dependent changes in quaternary structure, interactions with hydrophobic molecules near the extracellular membrane surface, and/or association with additional protein partners.

[1] Department of Molecular & Cell Biology, University of California, Berkeley, Berkeley, CA 94720, USA. [2] Helen Wills Neuroscience Institute, University of California, Berkeley, Berkeley, CA 94720, USA. [3] California Institute for Quantitative Biology (QB3), University of California, Berkeley, Berkeley, CA 94720, USA. [4] These authors contributed equally: Baobin Li, Christopher M. Hoel. ✉email: brohawn@berkeley.edu

 1

Tweety homologs (TTYHs) are conserved transmembrane proteins present in all eukaryotes including three members (TTYH1, TTYH2, and TTYH3) in humans[1–5]. The founding member, *Drosophila melanogaster* tweety (TTY), was named after a flightless cartoon bird due to its presence in the *flightless* genomic locus where mutations result in loss of flying ability[6,7], though ablation of the *tty* gene without concurrent loss of the *flightless* gene does not result in this phenotype. TTYHs have been reported to form $Ca^{2+}$- and cell volume-regulated anion channels and to contribute to cell adhesion, migration, and developmental signaling. In addition, TTYHs have been implicated in multiple diseases including cancers. Still, the structures, functions, and physiological roles of TTYHs have remained unclear.

Physiologically, TTYH2 and TTYH3 are broadly expressed, while expression of TTYH1 is primarily limited to the nervous system, testes, and stem cells[4,8–10]. Dysregulation of TTYHs expression has been implicated in some cancers: TTYH2 is upregulated in renal cell carcinoma[2], colon carcinomas[11], and osteosarcomas[12]; TTYH1 is expressed in gliomas where it contributes to brain colonization[13] and a TTYH1-C19MC microRNA cluster genetic fusion causes embryonal tumors with multilayered rosettes[10]; and TTYH3 is upregulated in gastric cancer with higher expression correlated with poor clinical outcomes[14] and a TTYH3-BRAF genetic fusion causes glioblastoma multiforme[15]. TTYHs have also been implicated in other pathologies. TTYH2 in myeloid cells interacts with SARS-CoV-2 Spike and may contribute to ACE2-independent immune cell infection and immune hyperactivation in severe cases of COVID-19[16]. TTYH1 is upregulated following epileptic events in central neurons and glia[17,18] and following inflammation in peripheral sensory neurons where it contributes to nociception and pain sensitization[19].

TTYHs are most often described as $Ca^{2+}$- and/or volume-regulated anion channels[5]. However, the evidence in support of channel activity to date is limited to five reports with inconsistent conclusions[3,20–23]. In the first study, TTYH2 and TTYH3 were reported to generate calcium-activated anion channels and TTYH1 was reported to generate a large conductance, calcium-independent volume-regulated anion channel[3,24]. A second study reported TTYH1 generates $Ca^{2+}$-dependent and volume-regulated anion channels[20]. A third study reported TTYH2 generates β-COP-regulated anion currents in low internal ionic strength and ascribed endogenous anionic currents in LoVo cancer cells to TTYH2[21]. A fourth study attributed endogenous anionic currents in SNU-601, HepG2, and LoVo cancer cells to TTYH2 and/or TTYH1 channel activity[23]. Finally, a fifth study reported TTYH1-3 form AQP4-dependent, $Ca^{2+}$-insensitive volume-regulated anion channels activated by cell swelling in astrocytes or in cultured cells upon cotransfection[22]. Among these studies, two reports identified the small molecule DIDS as a channel blocker[3,20] and three reported point mutations with modest effects on ion selectivity ($TTHY1_{R371Q}$, $TTYH3_{H370D}$, and $TTYH3_{R367Q}$)[3,20], DIDS block ($TTHY1_{F394S}$)[20], or current magnitude ($TTYH1_{R165A/R165C}$)[22] (numbering refers to human sequences).

Other work has implicated TTYHs in aspects of cell migration, cell adhesion, and developmental signaling. TTYH2 expression in two cell lines results in increased cell aggregation[11]. TTYH1 is implicated in maintenance of neural stem cells through positive feedback between TTYH1 and Notch signaling[25,26]. Heterologous expression of TTYH1 in cultured cells induced growth of filopodia[4,17] in which TTYH1 colocalized with α5 integrin[4]. Endogenous TTYH1 was localized to neurites in rat hippocampal neurons[4,17] and to extending processes of glia[18]. In cancer cells of glial origin, TTYH1 localized to tips of tumor microtubes, neurite-like long membranous extensions involved in proliferation, invasion, and network formation among glioma cells[13]. Finally, knockdown of TTYH1 within the tumor cells resulted in altered morphology of cellular protrusions, resulting in "beading" of neurites and glioma tumor microtubes, and reduction of tumor growth and improved survival in vivo[13,17].

Intrigued by reports that TTYHs form $Ca^{2+}$- or volume-regulated anion channels in the absence of homology to known ion channels, we set out to characterize TTYH structure and function. Here, we present cryo-EM structures of TTYH2 and TTYH3 and a functional analysis of TTYH2. We conclude TTYHs are not pore-forming subunits of ion channels, but adopt a previously unobserved fold capable of $Ca^{2+}$-dependent changes in oligomeric state and quaternary structure.

## Results

Full-length *Mus musculus* TTYH2 was expressed in HEK293T cells with a cleavable C-terminal fusion to EGFP, extracted and purified in detergent, and reconstituted into lipid nanodiscs formed by the scaffold protein MSP1E3D1 and lipids in the presence of 1 mM $Ca^{2+}$ (Supplementary Fig. 1). We determined the cryo-EM structure of TTYH2 in the presence of $Ca^{2+}$ to 3.3 Å resolution (Fig. 1, Supplementary Fig. 2). The map enabled de novo modeling of 396 of 532 amino acids from each 59 kDa TTYH2 protomer chain with the majority of unmodeled residues within the unstructured C-terminal region (amino acids 415–532) and others within poorly resolved segments of the N-terminus and intracellular loops. TTYH2 adopts a previously unobserved fold among proteins of known structure; Dali[27] searches using intact TTYH2 or isolated transmembrane or extracellular regions against the protein structure database do not return high confidence structural homologs.

In the presence of calcium, TTYH2 adopts a side-by-side homodimer (a "*cis*-dimer") within a single nanodisc membrane (Fig. 1). Each protomer consists of five transmembrane helices (TM1–5) and an extracellular domain (ED) which extends 75 Å from the membrane surface between an extracellular N-terminus and intracellular C-terminus. The ED largely consists of a kinked bundle of four helices (EDH1–4). Two N-linked glycosylation sites are observed and partially modeled at N129 on EDH1 and N352 on EDH4. EDH1 is a relatively unbroken helix extending from TM2, EDH2 and EDH4 are split into two segments each (a and b) by helical breaks, and EDH3 is broken into four segments (a–d) with EDH3b and EDH3c forming a short helix-turn-helix motif positioned with their helical axes approximately perpendicular to those of EDH3a and EDH3d. The ED appears rigidly packed and is stabilized above and below EDH3b and EDH3c by disulfide bonds between EDH3d and EDH4a (between C300 and C367) and EDH3a and EDH4b (between C274 and C382). The TTYH2 protomer architecture is in close accordance with one investigation of TTYH topology[28] and in contrast to other predictions[3,22,24].

The high degree of homology between TTYHs (*M. musculus* TTYH1-3 display ~40% identity across their sequence), suggests a common molecular architecture. To test this, we investigated the structure of *M. musculus* TTYH3. TTYH3 was expressed, purified, and reconstituted into lipid nanodiscs in the presence of 1 mM $Ca^{2+}$ similarly to TTYH2 (Supplementary Fig. 1). A cryo-EM reconstruction of TTYH3 was determined to 3.2 Å resolution and the de novo modeled structure was compared to TTYH2 (Supplementary Fig. 3). The overall protomer architecture and *cis*-dimerization arrangement was nearly identical between TTYH2 and TTYH3 (overall r.m.s.d. = 1.4 Å). The most substantial difference is the absence of TM1 from one protomer in TTYH3: we found during data processing that TTYH3 reconstructions were clearly asymmetric with thinner nanodisc density

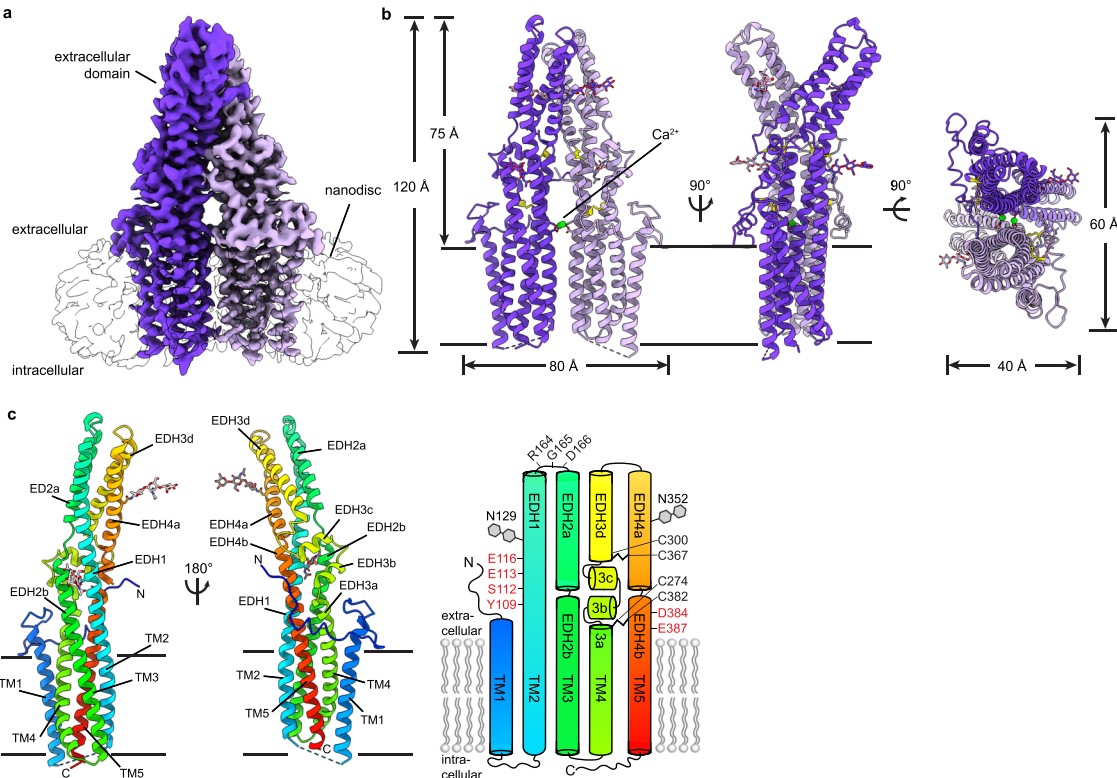

**Fig. 1 Structure of a TTYH2 *cis*-dimer in the presence of Ca²⁺.** **a** Cryo-EM map of a TTYH2 *cis*-dimer in MSP1E3D1 nanodiscs in the presence of Ca²⁺ at 3.3 Å resolution viewed from the membrane plane. Density from the TTYH2 protomers is colored dark and light purple and the nanodisc is white. **b** Model of the TTYH2 *cis*-dimer viewed from the membrane and extracellular side. N-linked glycosylation sites and disulfide bonds in the extracellular domain are drawn as sticks and Ca²⁺ ions are shown as green spheres. **c** A TTYH2 protomer viewed from the membrane plane and cartoon illustration of domain topology with rainbow coloring from blue (N-terminus) to red (C-terminus). Positions of disulfide bonds, glycosylation sites, and residues discussed in the text are indicated.

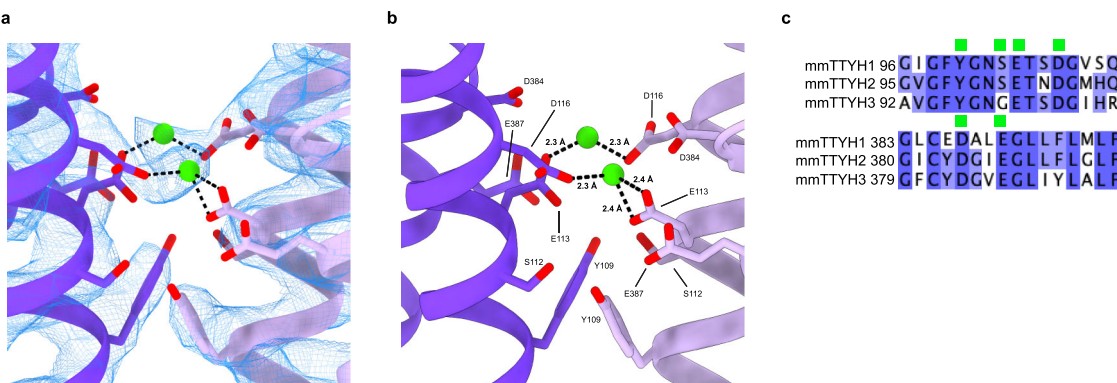

**Fig. 2 Extracellular Ca2⁺ binding sites in TTYH2. a**, **b** View from the membrane plane of the TTYH2 *cis*-dimer model (purples) and cryo-EM density (blue mesh). Electronegative side chains from each protomer are displayed as sticks and Ca²⁺ ions are shown as green spheres. **c** Alignment of mouse TTYHs with electronegative residues near the Ca²⁺ binding sites indicated with green boxes and conservation colored from white (not conserved) to blue (most conserved).

on one side of the complex. This thinner disc density displayed weak or absent density for TM1. While a subset of particles in the TTYH2 dataset were similarly asymmetric (Supplementary Fig. 3d), most displayed symmetric and interpretable density for TM1s in each protomer. Whether this difference is functionally relevant remains to be determined. We focus our discussion on the TTYH2 structure unless otherwise noted.

Association between TTYH2 protomers in the *cis*-dimer buries an interface of 1556 Å² and involves residues in the transmembrane and EDs, primarily in TM2, TM5, EDH1, EDH3d, EDH4a,

and EDH4b. A striking feature of the interface is the close juxtaposition of a surface formed by conserved negatively charged or electronegative residues from each protomer in the ED just above the membrane surface (Fig. 2a–c). This region includes residues from EDH1 (Y109, S112, E113, and E116) and EDH4b (D384 and E387) with Y109 and S112 underneath E113, E116, D384, and E387. The cryo-EM map contains strong density in the space between the charged residues bridging opposing protomers (Fig. 2a). Based on electrostatic and geometric features of this site, we speculate that these densities correspond to coordinated Ca²⁺

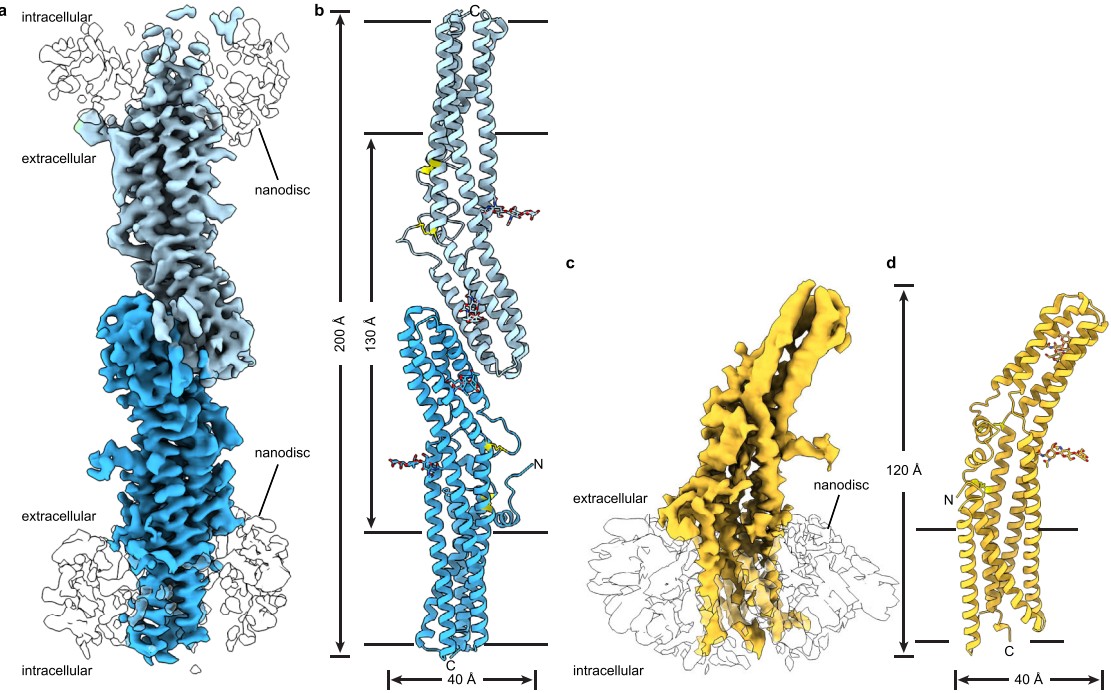

**Fig. 3 Structures of a TTYH2 *trans*-dimer and monomer in the absence of Ca²⁺. a** Cryo-EM map of a TTYH2 *trans*-dimer in MSP1E3D1 nanodiscs in the absence of Ca²⁺ at 3.9 Å resolution viewed from the membrane plane. Density from the TTYH2 protomers is colored dark and light blue and the nanodiscs are white. **b** Model of the TTYH2 *trans*-dimer viewed from the membrane. N-linked glycosylation sites and disulfide bonds in the extracellular domain are drawn as sticks. **c** Cryo-EM map of a TTYH2 monomer in MSP1E3D1 nanodiscs in the absence of Ca²⁺ at 4.0 Å resolution viewed from the membrane plane. Density from the TTYH2 protomer is colored orange and the nanodisc is white. **d** Model of the TTYH2 monomer viewed from the membrane. N-linked glycosylation sites and disulfide bonds in the extracellular domain are drawn as sticks.

ions that would reduce repulsion otherwise expected to occur between the electronegative surfaces of each protomer.

We reasoned that if Ca²⁺ was an important component of the *cis*-dimer interface for bridging the electronegative surfaces of each protomer, removing Ca²⁺ from TTYH2 could alter its oligomeric state. To test this, we purified and reconstituted TTYH2 into lipid nanodiscs without Ca²⁺ and in the presence of EGTA to maintain low free Ca²⁺ concentrations for structure determination. We also pretreated filter paper used for EM grid blotting with EGTA to sequester the high residual Ca²⁺ known to be present in this material to prevent Ca²⁺ transfer to the TTYH2 nanodisc sample[29].

The Ca²⁺-free TTYH2 cryo-EM data did reveal a change in oligomeric state, but surprisingly we identified two different conformations in approximately equal proportions: the expected monomeric TTYH2 (to 4.0 Å resolution) and an unexpected head-to-head homodimer (a "*trans*-dimer" to 3.9 Å resolution) in which TTYH2 protomers embedded in separate nanodiscs associate via a buried surface at the distal end of each ED (Fig. 3, Supplementary Fig. 4). The *trans*-dimer therefore bridges two opposing membranes over a ~130 Å extracellular space. Protomers within monomeric, *trans*-dimeric, and *cis*-dimeric TTYH2 adopt overall very similar conformations (pairwise protomer r.m.s.d. = 1.3–1.5 Å) with conformational changes largely limited to subtle rearrangements of side chains (Supplementary Fig. 5). The *trans*-dimeric interface is smaller than the *cis*-dimeric interface, burying 908 Å² compared to 1556 Å² and is largely polar with eight intersubunit hydrogen bonds (between residues D166-Q316, D166-T320, Q169-T321, and Q325-Q325), π-π stacking interactions (between F173-F173), and nonpolar interactions (between I324-L329). The *trans*- and *cis*-dimeric

interfaces are partially overlapping with residues Q316, R317, T320, and T321 each contributing to both TTYH2 *cis*-dimer and *trans*-dimer interactions (Fig. 4). As a consequence of this partial overlap in interfaces, *cis*-dimerization and *trans*-dimerization are expected to be mutually exclusive. Notably, residues involved both *cis*- and *trans*-dimerization interfaces are highly conserved with markedly lower conservation evident on the opposing face of the molecule that is not involved in oligomerization (Fig. 4).

A striking aspect of the TTYH structures is that they display no obvious path for ion conduction across the membrane as would be expected for an anion channel. One possible location for an ion conducting pore is within each protomer, as observed in other anion channels including CLCs[30]. In both TTYH2 and TTYH3, TM2, TM3, TM4, and TM5 pack tightly through a predominantly hydrophobic interface leaving no path for conduction in the absence of dramatic conformational changes. A second possible location for a pore is between subunits of a *cis*-dimer. In TTYH2 and TTYH3, this interface is less tightly packed, but almost exclusively hydrophobic, which would create a high energy barrier for ion passage. A third possible route for ion conduction is along hydrophilic grooves in the transmembrane region facing and partially exposed to the lipid bilayer, similar to those in TMEM16 family anion channels[31,32]. TTYH2 and TTYH3 display no such hydrophilic groove and rather show a hydrophobic surface across the membrane typical of most membrane proteins. In addition to the lack of a clear pore in the present structures, mapping point mutations previously reported to alter channel properties[3,20,22] onto the structures shows that these residues are unlikely to contribute to putative conduction paths: two are near the top of the EDH4b and contribute or are close to the *cis*-dimer interface (R368 and H371), one is membrane facing at the

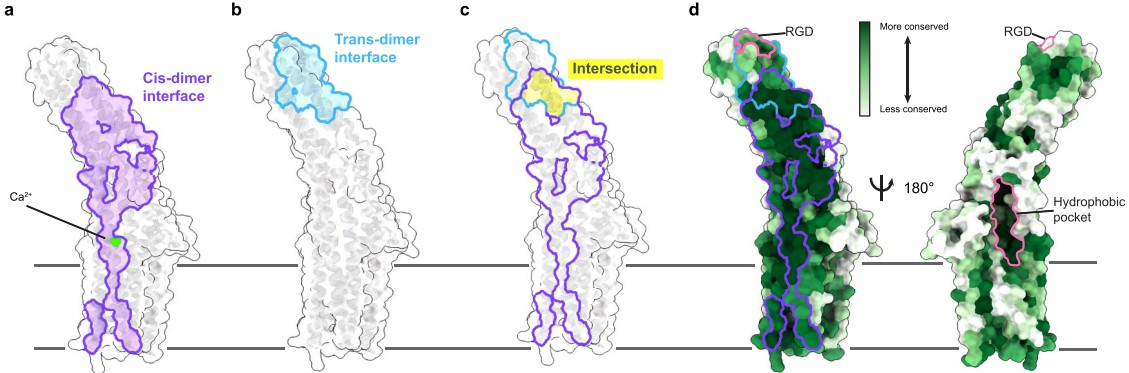

**Fig. 4 Conservation and partial overlap of the TTYH2 *cis*- and *trans*-dimerization interfaces. a** "Open book" view of the molecular surface of one TTYH2 protomer. The *cis*-dimerization interface is outlined and colored purple. The position of the bound $Ca^{2+}$ is shown. **b** Open book view of the molecular surface of one protomer from the TTYH2 *trans*-dimer. The *trans*-dimerization interface is outlined and colored blue. **c** Overlay of *cis*-dimerization (purple outline) and *trans*-dimerization (blue outline) interfaces with intersection highlighted in yellow. **d** TTYH2 surface colored according to conservation among chordate TTYH1-3 sequences. *cis*-dimer and *trans*-dimer interfaces are outlined as in **a–c**. The extracellular hydrophobic pocket and RGD motif are outlined in pink.

extracellular end of TM5 (F391), and one is present on the distal tip of the ED in the short linker between EDH1 and EDH2a (R164) (Supplementary Fig. 6).

To further investigate potential ion channel activity in TTYH2, we performed electrophysiological recordings from TTYH2-expressing cells (Supplementary Fig. 6). Consistent with the structural analysis, we were unable to observe the previously described TTYH2-dependent currents[3,20–23]. Cells expressing TTYH2 and AQP4 did not display currents significantly different from control cells under basal conditions or after perfusion of hypotonic extracellular solution that resulted in cell swelling (Supplementary Fig. 6). Based on structural data showing the lack of putative conduction paths in TTYH2 and TTYH3 and these electrophysiological results, we conclude TTYH proteins are not pore-forming subunits of anion channels.

## Discussion

TTYHs are most often described as anion channels regulated by $Ca^{2+}$ or cell swelling[5]. However, there are only limited experimental data that support this assignment[3,20–23], some results from these studies are contradictory, and other evidence has been presented[33,34] to refute the conclusion reached in the first of these reports that TTYH1 generates maxi-anion channels upon heterologous expression in cultured cells[3]. Evidence presented here is inconsistent with TTYHs forming pore-forming subunits of anion channels. There are no apparent pathways for ion transport across the membrane within a TTYH protomer, along the outside of protomers through hydrophilic grooves, or between protomers in *cis*-dimers and we are unable to observe TTYH2 currents in cells. Although it is possible that stimuli other than $Ca^{2+}$ or hypotonicity-induced cell swelling opens pores in TTYHs, the structural evidence leads us to hypothesize TTYHs perform other cellular functions. It may be that prior reports of TTYH-dependent channel activity[3,20–23] are a consequence of indirect effects of TTYH expression or manipulation on other ion channels expressed in cells including volume-regulated anion channels formed by LRRC8A-E. It also remains a possibility that TTYHs serve as auxiliary subunits of other ion channels. Either of these scenarios could plausibly explain prior results including the observation that volume-regulated anion channel currents are completely ablated in TTYH1 and/or TTYH2 knockout or knockdown cells and rescued by subsequent reintroduction of TTYH1 or TTYH2[22,23].

How might we explain the physiological and pathological role of TTYHs if not through the direct or indirect conduction of anions? A notable structural feature in the ED of TTYHs suggests that interactions with lipids or other hydrophobic ligands could play a functional role. Within each TTYH protomer, we observe a large, conserved, and primarily hydrophobic pocket facing away from the *cis*-dimer interface (Fig. 4d, Supplementary Fig. 7). The pocket is positioned just above the outer membrane leaflet and is open both to the bilayer and extracellular solution. In TTYH2 *cis*-dimers, we observe horseshoe-shaped density within the pocket, although the density is not sufficiently featured at this resolution to unambiguously identify and model (Supplementary Fig. 7). The pocket and unassigned density is similarly observed in TTYH3 *cis*-dimers, TTYH2 monomers, and TTYH2 *trans*-dimers, suggesting it may be a conserved site for binding small molecules or extended and primarily apolar lipid headgroups. It is conceivable that TTYHs utilize this pocket for binding or transporting lipids, other ligands, or short peptides from interacting proteins.

TTYHs including TTYH2 have also been implicated in cell development, growth, and adhesion. Could these roles be related to functions other than anion conduction? Shortly after its discovery, it was recognized that TTYH2 contains an RGD sequence motif[2,35]. Similar motifs in cell matrix proteins mediate direct interactions with RGD-binding integrins and RGD-containing peptides and other RGD mimetics are sufficient for integrin binding[36,37]. Consistent with this notion, TTYH1 was reported to colocalize by immunofluorescence with α5 integrin in the filopodia of transfected cells[4]. In mouse TTYH2, the RGD motif (R164, G165, D166) is positioned at the distal tip of the ED within the short linker between helices EDH1 and EDH2 (Fig. 4d, Supplementary Fig. 8). The RGD sequence is well conserved among TTYH2s. TTYH1 and TTYH3 sequences generally maintain positive and negative charges at positions corresponding to R164 and D166 and display more variation in the identity of the central residue. Intriguingly, one of the point mutations previously reported to impact TTYH2 ion channel activity and proposed to form part of the ion conduction pore was of the arginine in this sequence (R164)[22]. Given that this residue is ~75 Å from the membrane surface, it may more likely be involved in protein-protein interactions (Fig. 4d, Supplementary Fig. 8). Indeed, the RGD motif forms part of the *trans*-dimerization interface in TTYH2. In contrast, it is exposed in TTYH2 *cis*-dimers and monomers with ~250 Å² of solvent accessible area

presented to solution by each RGD. This difference suggests that if RGD or RGD-like motifs in TTYHs do interact with integrins, the interaction could be regulated by extracellular $Ca^{2+}$ levels: cis-dimers favored in the presence of $Ca^{2+}$ would display RGDs available for interaction, while trans-dimerization in low $Ca^{2+}$ conditions would disfavor integrin interaction by sequestration of the RGD surface. It may therefore be that the reduction in current previously observed as a result of mutating R164[22] is a consequence of disrupting TTYH trans-dimerization or TTYH-integrin interactions that in turn impact cell swelling-activated channel activity.

During preparation of this manuscript, an independent report of TTYH structures (H. sapiens TTYH1-3) was published[38]. In that report, human TTYH1, TTYH2, and TTYH3 structures are captured as cis-dimers similar to mouse TTYH2 and TTYH3 cis-dimers reported here (Supplementary Fig. 5, pairwise protomer r.m.s.d. = 1.2 Å). Structural features including the $Ca^{2+}$ interaction site bridging protomers in the cis-dimer and the presence of a hydrophobic pocket in the ED with unmodeled density features are similarly conserved. Notably, reported human TTYH1 and TTYH2 structures determined in $Ca^{2+}$-free conditions are also cis-dimers, while we observe monomers and trans-dimers of mouse TTYH2 prepared in $Ca^{2+}$-free conditions here. It may be that human and mouse TTYH2 have a different propensity for $Ca^{2+}$-dependent changes in oligomerization state. However, human and mouse TTYH2 display a high degree of overall sequence (83% identity) and structural conservation, including at the trans-dimerization interface where 18/19 residues involved are identical between human and mouse. The only difference (mouse G327 and human A327) is not expected to impact the propensity of TTYH2 to trans-dimerize. We therefore do not expect sequence differences between species at the interface to account for the difference in observed oligomeric state. Alternatively, sufficient $Ca^{2+}$ may have been present in the reported $Ca^{2+}$-free human TTYH1 and TTYH2 samples (perhaps due to leaching from filter paper) to occupy $Ca^{2+}$ sites and promote cis-dimerization[29].

We observed $Ca^{2+}$-dependent changes in TTYH2 oligomerization state in structures determined at 1 mM $Ca^{2+}$ and in the absence of $Ca^{2+}$. Whether these oligomeric state changes are physiologically relevant depends on several currently unknown factors. For dissociation of cis-dimers into monomers, these include: (1) the affinity of TTYHs for $Ca^{2+}$, (2) the related dependence of TTYH oligomeric state changes on $[Ca^{2+}]_{ext}$, and (3) the range of $[Ca^{2+}]_{ext}$ experienced by TTYH-expressing cells under different conditions. For association of monomers into trans-dimers, these additionally include: (4) the binding constant for trans-dimerization and (5) the surface density of natively expressed TTYHs. Future studies into these factors and potential interactions between TTYHs and integrins or other extracellular binding partners (and their $Ca^{2+}$-dependence) will be important for interpreting the physiological and functional roles of the TTYH structures reported here.

## Methods

**Cloning, expression, and purification.** Genes encoding M. musculus TTYH2 and TTYH3 were codon-optimized for expression in H. sapiens embryonic kidney cells (HEK293S GNTI− cells) and S. frugiperda cells (Sf9 cells), respectively, and synthesized (Integrated DNA Technologies) (Supplementary Table 1). The TTYH2 gene was cloned into a custom vector based on the pEG Bacmam[39] backbone with an added C-terminal PreScission protease (PPX) cleavage site, linker sequence, superfolder GFP (sfGFP) and 10×His tag, generating a construct for expression of TTYH2-SNS-LEVLFQGP-TAAAA-sfGFP-GGG-10×His. The construct was transformed into the DH10Bac E. coli strain to generate the recombinant bacmid DNA which was then used to transfect insect cells to generate BacMam P1 virus. P2 virus was then generated by infecting cells at two million cells per mL with P1 virus at a multiplicity of infection of roughly 0.1, with infection monitored by fluorescence and harvested after 72 h. P3 virus was generated in a similar manner.

P3 viral stock was then used to infect suspension HEK293S GNTI- cells at two million cells per mL grown at 37 °C with 5% $CO_2$. After 24 h, 10 mM sodium butyrate was added and cells were grown for another 48 h at 30 °C in the presence of 5% $CO_2$. Cells were pelleted, flash-frozen in liquid nitrogen, and stored at −80 °C.

A cell pellet from 2 L culture was thawed on ice and lysed in 100 mL of hypotonic buffer containing 10 mM Tris pH 8.0, 1 mM $CaCl_2$, 1 mM phenylmethylsulfonyl fluoride, 1 mM E64, 1 mg/mL pepstatin A, 10 mg/mL soy trypsin inhibitor, 1 mM benzimidine, 1 mg/mL aprotinin, and 1 mg/mL leupeptin. The membrane fraction was collected by centrifugation at 150,000 × g for 45 min and homogenized with a Dounce homogenizer in 100 mL lysis buffer containing 20 mM Tris pH 8.0, 150 mM NaCl, 1 mM $CaCl_2$, 1 mM phenylmethylsulfonyl fluoride, 1 mM E64, 1 mg/mL pepstatin A, 10 mg/mL soy trypsin inhibitor, 1 mM benzimidine, 1 mg/mL aprotinin, 1 mg/mL leupeptin, 10 μL Benzonase® endonuclease (EMD Millipore), 1% n-dodecyl-b-D-maltopyranoside (DDM, Anatrace, Maumee, OH), 0.2% cholesteryl hemisuccinate tris salt (CHS, Anatrace). Protein was extracted with gentle stirring for 2 h at 4 °C. Five milliliters of Sepharose resin coupled to anti-GFP nanobody was added to the supernatant and stirred gently for 2 h at 4 °C. The resin was collected in a column and washed with 50 mL buffer 1 (20 mM Tris, 500 mM NaCl, 1 mM $CaCl_2$, 0.025% DDM, 0.005% CHS, pH 8.0), 150 mL buffer 2 (20 mM Tris, 150 mM NaCl, 1 mM $CaCl_2$, 0.025% DDM, 0.005% CHS, pH 8.0), and 20 mL of buffer 1. PPX (~0.5 mg) was added into the washed resin in 5 mL buffer 2 and rocked gently overnight. Cleaved TTYH2 was eluted and concentrated to ~0.5 mL with an Amicon Ultra spin concentrator (50 kDa cutoff, MilliporeSigma, USA). The concentrated protein was subjected to size exclusion chromatography using a Superose 6 Increase 10/300 column (GE Healthcare, Chicago, IL) run in buffer 3 (20 mM Tris pH 8.0, 150 mM NaCl, 1 mM $CaCl_2$, 0.025% DDM, 0.005% CHS) on a NGC system (Bio-Rad, Hercules, CA). The peak fractions were collected, and spin concentrated for nanodisc reconstitution. For cryo-EM samples in absence of $Ca^{2+}$, 5 mM EGTA was added into the buffer instead of 1 mM $CaCl_2$.

The TTYH3 gene was cloned into a custom vector based on the pACEBAC1 backbone (MultiBac, Geneva Biotech) with an added C-terminal PreScission protease (PPX) cleavage site, linker sequence, superfolder GFP (sfGFP) and 7×His tag, generating a construct for expression of TTYH3-SNS-LEVLFQGP-SRGGSGAAAGSGSGS-sfGFP-GSS-7×His in Sf9 cells. The construct was transformed into the DH10MultiBac E. coli strain to generate recombinant bacmid DNA which was then used to transfect Sf9 to generate P1 virus. P2 virus was generated by infecting Sf9 cells at two million cells per mL with P1 and harvested after 72 h. P3 virus was generated in a similar manner. P3 viral stock was used to infect suspension Sf9 cells at two million cells per mL grown at 27 °C. After 72 h, cells were pelleted, flash-frozen in liquid nitrogen, and stored at −80 °C. A cell pellet from 1 L culture was thawed on ice and cells were lysed by sonication in 100 mL buffer (10 mM Tris pH 8.0, 1 mM $CaCl_2$, 1 mM phenylmethylsulfonyl fluoride, 1 mM E64, 1 mg/mL pepstatin A, 10 mg/mL soy trypsin inhibitor, 1 mM benzimidine, 1 mg/mL aprotinin, and 1 mg/mL leupeptin). The membrane fraction was collected by centrifugation at 150,000 × g for 45 min, and residual nucleic acid was removed from the top of the membrane pellet using Dulbecco's phosphate buffered saline (DPBS). The membrane was homogenized with a Dounce homogenizer in 100 mL lysis buffer (20 mM Tris pH 8.0, 150 mM NaCl, 1 mM $CaCl_2$, 1 mM phenylmethylsulfonyl fluoride, 1 mM E64, 1 mg/mL pepstatin A, 10 mg/mL soy trypsin inhibitor, 1 mM benzimidine, 1 mg/mL aprotinin, 1 mg/mL leupeptin, 10 μL Benzonase® endonuclease (EMD Millipore), 1% n-dodecyl-b-D-maltopyranoside (DDM, Anatrace, Maumee, OH), 0.2% cholesteryl hemisuccinate tris salt (CHS, Anatrace)). Protein was extracted with gentle stirring for 2 h at 4 °C. 5 mL Sepharose resin coupled to anti-GFP nanobody was added to the supernatant and stirred gently for 2 h at 4 °C. The resin was collected in a column and washed with 50 mL buffer 1 (20 mM Tris, 500 mM NaCl, 1 mM $CaCl_2$, 0.025% DDM, 0.005% CHS, pH 8.0), 150 mL buffer 2 (20 mM Tris, 150 mM NaCl, 1 mM $CaCl_2$, 0.025% DDM, 0.005% CHS, pH 8.0), and 20 mL of buffer 1. PPX (~0.5 mg) was added into the washed resin in 5 mL buffer 2 and rocked gently overnight. Cleaved TTYH3 was eluted and concentrated to ~0.5 mL with an Amicon Ultra spin concentrator (50 kDa cutoff, MilliporeSigma, USA). The concentrated protein was subjected to size exclusion chromatography using a Superose 6 Increase 10/300 column (GE Healthcare, Chicago, IL) run in buffer 3 (20 mM Tris pH 8.0, 150 mM NaCl, 1 mM $CaCl_2$, 0.025% DDM, 0.005% CHS) on a NGC system (Bio-Rad, Hercules, CA). The peak fractions were collected and spin concentrated for nanodisc reconstitution[40].

**Nanodisc reconstitution.** Freshly purified proteins were reconstituted into MSP1E3D1 nanodiscs with a 2:1:1 DOPE:POPS:POPC lipid mixture (mol:mol, Avanti, Alabaster, AL) at a final molar ratio of TTYH:MSP1E3D1:lipid of 1:4:400. Lipids in chloroform were mixed, dried under argon, washed with pentane, dried under argon, and dried under vacuum overnight in the dark. Dried lipids were rehydrated in buffer 4 containing 20 mM Tris pH 8.0, 150 mM NaCl, 1 mM $Ca^{2+}$ and clarified by bath sonication. DDM was added to a final concentration of 8 mM. Proteins were mixed with lipids and incubated at 4 °C for 30 min before addition of MSP1E3D1 protein. After incubation for 10 min at 4 °C, 100 mg of Biobeads SM2 (Bio-Rad, USA) (prepared by sequential washing in methanol, water, and buffer 4 and weighed damp following bulk liquid removal) was added and the mixture was

rotated at 4 °C overnight. The sample was spun down to facilitate removal of solution from the Biobeads and reconstituted TTYHs further purified on a Superose 6 increase column. The peak fractions were collected and spin concentrated (50 kDa MWCO) to 1.0–1.3 mg/mL for grid preparation. For cryo-EM samples in absence of $Ca^{2+}$, 5 mM EGTA was added into the buffer instead of 1 mM $CaCl_2$.

**Grid preparation.** The TTYH2 and TTYH3 nanodisc samples were centrifuged at $21,000 \times g$ for 10 min at 4 °C. A 3 µL sample was applied to holey carbon, 300 mesh R1.2/1.3 gold grids (Quantifoil, Großlöbichau, Germany) that were freshly glow discharged for 25 s. Sample was incubated for 5 s at 4 °C and 100% humidity prior to blotting with Whatman #1 filter paper for 3–3.5 s at blot force 1 and plunge-freezing in liquid ethane cooled by liquid nitrogen using a FEI Mark IV Vitrobot (FEI/Thermo Scientific, USA). For the $Ca^{2+}$-free condition, 5 mM EGTA was used in place of $Ca^{2+}$ in purification buffers and the filter paper was washed sequentially in water, 5 mM EGTA solution, and water before drying overnight under vacuum to remove the high concentrations of $Ca^{2+}$ in the filter paper that could otherwise be transferred to the sample[29].

**Cryo-EM data acquisition.** Grids were clipped and transferred to a FEI Talos Arctica electron microscope operated at 200 kV. Fifty frame movies were recorded on a Gatan K3 Summit direct electron detector in super-resolution counting mode with pixel size of 0.5685 Å for TTYH2 and TTYH3 with $Ca^{2+}$ and 0.5775 Å for TTYH2 in absence of $Ca^{2+}$. The electron dose rate was 9.556, 9.021, and 8.849 e$^-$ Å$^2$ s$^-$ and the total dose was 50.225, 47.28, and 50.0 e$^-$ Å$^2$ for TTYH2 with $Ca^{2+}$, TTYH3 with $Ca^{2+}$ and TTYH2 without $Ca^{2+}$, respectively. Nine movies were collected around a central hole position with image shift and defocus was varied from −0.6 to −1.8 µm through SerialEM[41]. See Table 1 for data collection statistics.

**Cryo-EM data processing.** A similar pipeline was followed to process all three datasets[40]. Motion-correction and dose-weighting were performed on all micrograph movies using RELION3.1's implementation of MotionCor2 and 2x "binned" from super resolution to physical pixel size[42–44]. CTFFIND-4.1 was used to estimate the contrast transfer function (CTF) parameters[45]. Micrographs were then manually sorted to eliminate subjectively bad micrographs, such as empty or contaminated holes. In addition, micrographs with a CTF maximum estimated resolution lower than 5 Å were discarded. Template-free auto-picking of particles was performed with RELION3.1's Laplacian-of-Gaussian filter yielding an initial set of particles. This initial set of particles underwent classification to generate templates, which were subsequently used for template-based auto-picking of particles. To "clean-up" these particles, they were first 2D-classified in both RELION3.1 and cryoSPARC v2[46,47], then iterative ab initio and heterogenous refinement jobs were used to isolate higher quality particles. After no further improvements in map quality were observed, the resulting subset of particles was used for training in Topaz[48]. The above "clean-up" steps were then repeated for the Topaz-picked particles until no further improvements were observed, at which point the Topaz picks were combined with the template-picked particles, duplicates removed, and input to Bayesian particle polishing in RELION3.1. The resulting "shiny" particles then underwent iterative homogenous and nonuniform refinements until no further improvement was observed. The initial resolution and dynamic mask nonuniform parameters were adjusted to yield the best performing refinement. UCSF pyem was used for conversion of files from cryoSPARC to Relion formats[49].

**Table 1 Cryo-EM data collection, refinement, and validation statistics.**

|  | TTYH2 *cis*-dimer with $Ca^{2+}$ (PDB 7RTT) (EMDB 24688) | TTYH3 *cis*-dimer with $Ca^{2+}$ (PDB 7RTW) (EMDB 24691) | TTYH2 *trans*-dimer without $Ca^{2+}$ (PDB 7RTU) (EMDB 24689) | TTYH2 monomer without $Ca^{2+}$ (PDB 7RTV) (EMDB 24690) |
|---|---|---|---|---|
| Data collection and processing |  |  |  |  |
| Magnification | ×36,000 | ×36,000 | ×36,000 | ×165,000 |
| Voltage (kV) | 200 | 200 | 200 | 200 |
| Micrographs (no.) | 5014 | 3217 | 7245 | 7245 |
| Electron exposure (e$^-$/Å$^2$) | 50.225 | 47.28 | 50 | 50 |
| Defocus range (µm) | −0.6 to −1.8 | −0.6 to −1.8 | −0.6 to −1.8 | −0.6 to −1.8 |
| Super-resolution pixel size (Å) | 0.5685 | 0.5685 | 0.5575 | 0.5575 |
| Map pixel size (Å) | 1.137 | 1.137 | 1.115 | 1.699 |
| Symmetry imposed | C2 | C1 | C1 | C1 |
| Topaz particle images (no.) | 4,220,298 | 1,475,741 | 1,380,683 | 2,351,846 |
| Final particle images (no.) | 96,342 | 96,285 | 135,683 | 186,697 |
| Map resolution (Å) | 3.34 | 3.23 | 3.89 | 3.96 |
| FSC threshold | 0.143 | 0.143 | 0.143 | 0.143 |
| Refinement |  |  |  |  |
| Initial model used (PDB code) | De novo | De novo | 7RTT | 7RTT |
| Model resolution (Å) | 3.5 | 3.5 | 3.8 | 4.1 |
| FSC threshold | 0.143 | 0.143 | 0.143 | 0.143 |
| Model composition |  |  |  |  |
| Nonhydrogen atoms | 6226 | 5630 | 5960 | 2880 |
| Protein residues | 791 | 699 | 754 | 366 |
| Ligands | 14 | 17 | 12 | 6 |
| *B* factors (Å$^2$) |  |  |  |  |
| Protein | 80.27 | 57.43 | 107.75 | 108.17 |
| Ligand | 96.52 | 57.70 | 110.82 | 193.51 |
| R.m.s. deviations |  |  |  |  |
| Bond lengths (Å) | 0.003 | 0.003 | 0.005 | 0.003 |
| Bond angles (°) | 0.544 | 0.566 | 0.692 | 0.727 |
| Validation |  |  |  |  |
| MolProbity score | 1.6 | 1.74 | 1.78 | 1.77 |
| Clashscore | 5.26 | 9.68 | 9.63 | 13.18 |
| Poor rotamers (%) | 0 | 0 | 1.26 | 0 |
| Ramachandran plot |  |  |  |  |
| Favored (%) | 95.40 | 96.53 | 96.78 | 97.24 |
| Allowed (%) | 4.6 | 3.47 | 3.22 | 2.76 |
| Disallowed (%) | 0 | 0 | 0 | 0 |

For the $Ca^{2+}$-free data, the above strategy was followed without knowledge of the *trans*-dimer state until its discovery during classification of the template-picked particles. After identification and separation of particles into monomeric and *trans*-dimeric classes, Topaz training and subsequent steps of the above pipeline were undertaken independently for each state (Supplementary Fig. 4).

For the TTYH2 with $Ca^{2+}$ data, C2 symmetry was applied for the final refinement. C2 symmetry was not applied in any other case. We note that a subset of TTYH2 with $Ca^{2+}$ particles yielded an asymmetric TTYH2 *cis*-dimer, similar to the map observed for TTYH3 (Supplementary Fig. 3). In addition, a small number of monomeric particles were observed, but were insufficient to generate a meaningful reconstruction.

**Modeling, refinement, and analysis**. Maps were sharpened using cryoSPARC, models were built de novo in Coot[50], real space refined in Phenix[51], and assessed for proper stereochemistry and geometry using Molprobity[52]. Phenix resolve density modification was applied to maps for visualization[53]. The N-terminus prior to TM1 was copied from AlphaFold2 predicted structures[54,55] and refined into the cryo-EM density. Structures were analyzed and figures were prepared with HOLE[56], DALI[27], PyMOL, ChimeraX[57], JalView[58], Prism 8, GNU Image Manipulation Program, and Adobe Photoshop and Illustrator software. Consurf[59] was used to map conservation onto the structure using an alignment of 689 TTYH1-3 sequences retrieved from the Pfam database.

**Electrophysiology**. We performed whole-cell recordings to study the proposed chloride channel function of TTYH2. The same TTYH2 construct used for structural studies was cotransfected with an AQP4-mCherry construct at a ratio of 1:1 into HEK293T cells with Fugene HD transfection reagent. The cells were grown in DMEM-F12 (Gibco) with 10% FBS, 2 mM L-glutamine, 100 units/mL penicillin and 100 μg/mL streptomycin at 37 °C and 5% $CO_2$. Whole-cell recordings were performed at room temperature ~24–48 h after transfection. Borosilicate glass pipettes were pulled to a resistance of 3–7 MΩ. An Axopatch 200B amplifier connected to a Digidata 1550B digitizer (Molecular Devices) was used for recording. The following ramp protocol was applied once every 15 s: $V_{hold} = -50$ mV; ramp from +100 mV to −100 mV, 1000 ms. The external bath solutions (isotonic buffer and hypotonic buffer) contained 70 mM Tris-HCl, 1.5 mM $CaCl_2$, 10 mM HEPES, and 10 mM glucose, 5 mM TEA-Cl, and 5 mM $BaCl_2$ adjusted to pH 7.3 with CsOH[22]. The osmolality of each solution was adjusted with sucrose: 100 mM sucrose for ~320 mOsm (for ISO) and 30 mM sucrose for ~240 mOsm (for HOS). Solution osmolality was measured with a vapor pressure osmometer (Vapro #5600, ELI-TechGroup). During recordings, cells were initially perfused with the isotonic buffer prior to perfusion with hypotonic buffer. The pipette solution contained 60 mM Tris-HCl, 70 mM Tris-Aspartic acid, 15 mM HEPES, 0.4 mM $CaCl_2$, 1 mM $MgCl_2$, 4 mM Mg-ATP, 0.5 mM Na-GTP, and 1 mM EGTA adjusted to pH 7.25 by CsOH. All data were acquired using Clampex 10.7 (Molecular Devices) and analyzed in Clampfit 10.7 (Molecular Devices) and GraphPad prism.

**Reporting summary**. Further information on research design is available in the Nature Research Reporting Summary linked to this article.

## Data availability

All data associated with this study will be publicly available. For the TTYH2 *cis*-dimer in the presence of $Ca^{2+}$, the final model is in the PDB under 7RTT, the final map is in the Electron Microscopy Data Bank (EMDB) under EMD-24688, and the original micrograph movies and final particle stack are in the Electron Microscopy Public Image Archive (EMPIAR) database under EMPIAR-10842. For the TTYH2 *trans*-dimer in the absence of $Ca^{2+}$, the final model is in the PDB under 7RTU and the final map is in the EMDB under EMD-24689. For the TTYH2 monomer in the absence of $Ca^{2+}$, the final model is in the PDB under 7RTV and the final map is in the EMDB under EMD-24690. Original micrograph movies and final particle stacks for the TTYH2 monomer and *trans*-dimer are in the EMPIAR database under EMPIAR-10843. For the TTYH3 *cis*-dimer in the presence of $Ca^{2+}$, the final model is in the PDB under 7RTW, the final map is in the EMDB under EMD-24691, and the original micrograph movies and final particle stack are in the EMPIAR database under EMPIAR-10850. Source data are provided with this paper.

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

# ARTICLE

30. Dutzler, R., Campbell, E. B., Cadene, M., Chait, B. T. & MacKinnon, R. X-ray structure of a ClC chloride channel at 3.0 Å reveals the molecular basis of anion selectivity. *Nature* **415**, 287–294 (2002).

31. Paulino, C., Kalienkova, V., Lam, A. K. M., Neldner, Y. & Dutzler, R. Activation mechanism of the calcium-activated chloride channel TMEM16A revealed by cryo-EM. *Nature* **552**, 421–425 (2017).

32. Dang, S. et al. Cryo-EM structures of the TMEM16A calcium-activated chloride channel. *Nature* **552**, 426–429 (2017).

33. Okada, Y. et al. The Puzzles of Volume-Activated Anion Channels. *Physiology and pathology of chloride transporters and channels in the nervous system Ch. 15.* 283–306 (Academic Press, 2010). https://doi.org/10.1016/b978-0-12-374373-2.00015-7.

34. Sabirov, R. Z. et al. The ATP-releasing maxi-Cl channel: its identity, molecular partners, and physiological/pathophysiological implications. *Life* **11**, 509 (2021).

35. He, Y., et al. The ubiquitin-protein ligase Nedd4-2 differentially interacts with and regulates members of the tweety family of chloride ion channels. *J. Biol. Chem.* **283**, 24000–24010 (2008).

36. Nagae, M. et al. Crystal structure of α5β1 integrin ectodomain: atomic details of the fibronectin receptor. *J. Cell Biol.* **197**, 131–140 (2012).

37. Ludwig, B. S., Kessler, H., Kossatz, S. & Reuning, U. RGD-binding integrins revisited: how recently discovered functions and novel synthetic ligands (re-) shape an ever-evolving field. *Cancers* **13**, 1711 (2021).

38. Sukalskaia, A., Straub, M. S., Deneka, D., Sawicka, M. & Dutzler, R. Cryo-EM structures of the TTYH family reveal a novel architecture for lipid interactions. *Nat. Commun.* **12**, 4893 (2021).

39. Goehring, A. et al. Screening and large-scale expression of membrane proteins in mammalian cells for structural studies. *Nat. Protoc.* **9**, 2574–2585 (2014).

40. Kern, D. M. et al. Cryo-EM structure of SARS-CoV-2 ORF3a in lipid nanodiscs. *Nat. Struct. Mol. Biol.* 1–10 (2021). https://doi.org/10.1038/s41594-021-00619-0.

41. Mastronarde, D. N. Automated electron microscope tomography using robust prediction of specimen movements. *J. Struct. Biol.* **152**, 36–51 (2005).

42. Zheng, S. Q. et al. MotionCor2: anisotropic correction of beam-induced motion for improved cryo-electron microscopy. *Nat. Methods* **14**, 331–332 (2017).

43. Zivanov, J., Nakane, T. & Scheres, S. H. W. A Bayesian approach to beam-induced motion correction in cryo-EM single-particle analysis. *IUCrJ* **6**, 5–17 (2019).

44. Zivanov, J. et al. New tools for automated high-resolution cryo-EM structure determination in RELION-3. *eLife* **7**, 163 (2018).

45. Rohou, A. & Grigorieff, N. CTFFIND4: fast and accurate defocus estimation from electron micrographs. *J. Struct. Biol.* **192**, 216–221 (2015).

46. Punjani, A., Rubinstein, J. L., Fleet, D. J. & Brubaker, M. A. cryoSPARC: algorithms for rapid unsupervised cryo-EM structure determination. *Nat. Methods* **14**, 290–296 (2017).

47. Punjani, A., Zhang, H. & Fleet, D. J. Non-uniform refinement: adaptive regularization improves single-particle cryo-EM reconstruction. *Nat. Methods* **17**, 1214–1221 (2020).

48. Bepler, T. et al. Positive-unlabeled convolutional neural networks for particle picking in cryo-electron micrographs. *Nat. Methods* **16**, 1153–1160 (2019).

49. Asarnow, D., Palovcak, E., Cheng, Y. UCSF pyem v0.5. *Zenodo* https://doi.org/10.5281/zenodo.3576630 (2019).

50. Emsley, P., Lohkamp, B., Scott, W. G. & Cowtan, K. Features and development of Coot. *Acta Crystallogr. Sect. D, Biol. Crystallogr.* **66**, 486–501 (2010).

51. Liebschner, D. et al. Macromolecular structure determination using X-rays, neutrons and electrons: recent developments in Phenix. *Acta Crystallogr. Sect. D, Struct. Biol.* **75**, 861–877 (2019).

52. Williams, C. J. et al. MolProbity: more and better reference data for improved all-atom structure validation. *Protein Sci.* **27**, 293–315 (2018).

53. Terwilliger, T. C., Ludtke, S. J., Read, R. J., Adams, P. D. & Afonine, P. V. Improvement of cryo-EM maps by density modification. *Nat. Methods* **17**, 923–927 (2020).

54. Jumper, J. et al. Highly accurate protein structure prediction with AlphaFold. *Nature* 1–11 (2021). https://doi.org/10.1038/s41586-021-03819-2.

55. Tunyasuvunakool, K. et al. Highly accurate protein structure prediction for the human proteome. *Nature* 1–9 (2021). https://doi.org/10.1038/s41586-021-03828-1.

56. Smart, O. S., Neduvelil, J. G., Wang, X., Wallace, B. A. & Sansom, M. S. HOLE: a program for the analysis of the pore dimensions of ion channel structural models. *J. Mol. Graph.* **14**, 354-60–354376 (1996).

57. Goddard, T. D. et al. UCSF ChimeraX: meeting modern challenges in visualization and analysis. *Protein Sci.* **27**, 14–25 (2018).

58. Waterhouse, A. M., Procter, J. B., Martin, D. M. A., Clamp, M. & Barton, G. J. Jalview version 2—a multiple sequence alignment editor and analysis workbench. *Bioinformatics* **25**, 1189–1191 (2009).

59. Ashkenazy, H. et al. ConSurf 2016: an improved methodology to estimate and visualize evolutionary conservation in macromolecules. *Nucleic Acids Res.* **44**, W344–W350 (2016).

## Acknowledgements

The authors thank J. Remis, D. Toso, and P. Tobias for microscope and computational support at the Cal-Cryo facility. The authors thank members of the Brohawn Laboratory for discussions and feedback on the project. S.G.B. is a New York Stem Cell Foundation-Robertson Neuroscience Investigator. This work was funded by the New York Stem Cell Foundation; NIGMS grant no. GM123496; a McKnight Foundation Scholar Award; a Sloan Research Fellowship; and a Winkler Family Scholar Award (to S.G.B.).

## Author contributions

B.L. generated expression constructs, purified proteins, prepared samples for cryo-EM, and performed electrophysiology experiments. B.L. and C.M.H. collected cryo-EM data, processed cryo-EM data, and determined and modeled structures. All authors analyzed data and wrote the manuscript. S.G.B. supervised the project.

## Competing interests

The authors declare no competing interests.
