## [Peer Review File · Nature Communications]

Structures of Tweety Homolog Proteins TTYH2 and TTYH3 reveal a Ca²⁺-dependent switch from intra- to inter- membrane dimerizationREVIEWER COMMENTS

Reviewer #1 (Remarks to the Author):

This manuscript by Li and colleagues reports cryo-EM structures of the mouse TTYH2&3 structures in the presence and absence of Ca²⁺. TTYH proteins have been proposed to function as volume-regulated anion channels. In the presence of Ca²⁺, the authors observed TTYH2&3 cis-dimers without an ion-conducting pore expected for an anion channel. In the absence of a different, a trans-dimer was obtained for TTYH2, with protomers from two membranes. The structural results were supported by electrophysiology measurements, which did not show ion conductance. The paper was clearly written. I recommend its publication in Nature Comm. With minor revision.

To disprove a hypothesis with mostly negative evidence is always challenging. The mouse TTYH cis-dimer structures agree with the recently published structures of human homologs, giving the reader additional confidence. However, in the SEC traces in 1 mM Ca²⁺ of both mouse TTYH2 and TTYH3 in nanodiscs (Figure S1a & c, right), there is a significant peak at the higher molecular side (~13 mL). Would the authors be willing to perform quick negative stain experiments of these larger peaks to rule out a higher oligomer with a pore?

The trans-dimer of TTYH2 observed in the absence of Ca²⁺ is novel, which may provide a structural basis for the cell adhesion function that the authors mentioned. However, the interface area of 908 Å² seems to be small for stable protein-protein interaction. Are there a lot of ion pairs and hydrogen bonds at the interface? Is it known that TTYHs express highly on the cell surface?

Minor comments:

1. ED refers to the extracellular domain in the text, whereas EDs1-4 refer to the four helices within the domain. Such a naming scheme is confusing. Perhaps, the authors could use EDHs1-4 for the latter.
2. In Figure S1c, mTTYH2 is probably mislabeled. Should it be mTTYH3?
3. In the Table for cryo-EM data collection and structure determination, the numbers of atoms should be written as 6,226, ...

Reviewer #2 (Remarks to the Author):

This manuscript by Li et al. reports the structures of Tweety (TTYH) 2 and 3 from mouse, and follows closely after a recently published paper from the Dutzler lab that reported similar structures of all three human homologs. The structures are (in terms of the monomer) essentially the same as reported by the Dutzler group. Similarly, Li et al. find dimers that associate through a large extracellular domain. The key finding here is that under very low calcium, the authors find a dimer between TTYH monomers in which each monomer is in a separate membrane. They offer a reasonable biochemical explanation for why the Dutzler group did not see this (invoking the well known 'stray' calcium from many sources).

The main value of this manuscript is the N=2 finding (i.e. both groups get the same structures, note the same extracellular hydrophobic cavity that may bind some yet to be discovered hydrophobic ligand, and importantly both groups provide evidence that these proteins are not chloride channels as previously proposed). Given this negation of a leading idea, it is important that a second study that independently arrived at the same conclusions be published. The authors however, should take the opportunity to show structural comparisons with the Dutzler structures (report RMSDs, etc.). While there are no surprises expected, such a comparison would add value.

The key new finding here is the trans-membrane dimer. The authors argue that one possible reason that this was not seen by Dutzler's group (apart from the stray calcium argument) is that there are

differences in the interface between the mouse and human proteins that could explain why one makes these dimers and the other does not. Given availability of the the structures of the human TTYHs, the authors are in a position to answer this question directly rather than speculate. They should check if there are sufficient differences in the sequences that make these surfaces that might support this idea. If so, such difference should be noted, and if not, that should also be noted as it would discount the idea that the sequence differences are the cause.

The key point about the trans-disc dimers that is not addressed is that such a low calcium environment is not common in extracellular spaces (or even in 'pre-extracellular spaces such as the ER lumen, Golgi lumen, etc. that serve as calcium stores). Hence, one wonders whether there are any conditions, or places, where TTHY might see this sort of low calcium environment. This point needs to be addressed as these interactions could be equivalent to fortuitous interactions that are made by proteins in protein crystals (i.e. surfaces that can come together under special conditions but that are not physiologically relevant). As the main novelty of this manuscript rests on this unexpected dimer (and it is so important that it is in the title), this point should be addressed clearly.

Point-by-point response

We would like to sincerely thank the reviewers for their careful reading and constructive feedback of the paper. We have addressed all comments raised in a revised manuscript and in the point-by-point response below. We hope you will agree the paper is substantially improved as a result.

Reviewer #1 (Remarks to the Author):

This manuscript by Li and colleagues reports cryo-EM structures of the mouse TTYH2&3 structures in the presence and absence of Ca²⁺. TTYH proteins have been proposed to function as volume-regulated anion channels. In the presence of Ca²⁺, the authors observed TTYH2&3 cis-dimers without an ion-conducting pore expected for an anion channel. In the absence of a different, a trans-dimer was obtained for TTYH2, with protomers from two membranes. The structural results were supported by electrophysiology measurements, which did not show ion conductance. The paper was clearly written. I recommend its publication in Nature Comm. With minor revision.

To disprove a hypothesis with mostly negative evidence is always challenging. The mouse TTYH cis-dimer structures agree with the recently published structures of human homologs, giving the reader additional confidence. However, in the SEC traces in 1 mM Ca²⁺ of both mouse TTYH2 and TTYH3 in nanodiscs (Figure S1a & c, right), there is a significant peak at the higher molecular side (~13 mL). Would the authors be willing to perform quick negative stain experiments of these larger peaks to rule out a higher oligomer with a pore?

We thank the reviewer for raising this important point. We analyzed the composition of this peak by SDS-PAGE and found it corresponds to TTYH2-free MSP1E3D1-liposome complexes or aggregates, excluding the possibility of larger TTYH2 oligomers in nanodiscs. This is now shown in the revised Figure S1A with the addition of a highlighted peak 1 in the chromatogram and corresponding Coomassie-stained gel band showing only MSP1E3D1 protein. We have observed similar peaks for membrane protein-free MSP1E3D1-liposome complexes or aggregates in reconstitutions of other similar-size targets (e.g. Kern et al. NSMB 2021, Li et al. Nature 2020, Reid et al. eLife 2020), consistent with it being a common side product of the nanodisc formation reaction.

The trans-dimer of TTYH2 observed in the absence of Ca²⁺ is novel, which may provide a structural basis for the cell adhesion function that the authors mentioned. However, the interface area of 908 Å² seems to be small for stable protein-protein interaction. Are there a lot of ion pairs and hydrogen bonds at the interface? Is it known that TTYHs express highly on the cell surface?

We have included a more detailed analysis of the trans-dimer interface to provide additional context for considering the physiological relevance of this interaction (see also response to Reviewer 2 below). Nineteen residues from each subunit contribute to a largely polar interaction surface that includes eight intersubunit hydrogen bonds (between D166-Q316, D166-T320, Q169-T321, and Q325-Q325), π-π stacking interactions (between F173-F173), and nonpolar interactions (between I324-L329). While the surface density of TTYH proteins on cells is not known, TTYHs are among the most highly expressed membrane proteins in some cells including glia based on mRNA sequencing (Nalamalapu et al. Front. Mol. Neurosci. 2021).

Minor comments:

1. ED refers to the extracellular domain in the text, whereas EDs1-4 refer to the four helices within the domain. Such a naming scheme is confusing. Perhaps, the authors could use EDHs1-4 for the latter.

We have changed the text and figures to use EDH for “extracellular domain helix”.

2. In Figure S1c, mTTYH2 is probably mislabeled. Should it be mTTYH3?

Yes, thank you. Corrected.

3. In the Table for cryo-EM data collection and structure determination, the numbers of atoms should be written as 6,226, ...

Corrected.

Reviewer #2 (Remarks to the Author):

This manuscript by Li et al. reports the structures of Tweety (TTYH) 2 and 3 from mouse, and follows closely after a recently published paper from the Dutzler lab that reported similar structures of all three human homologs. The structures are (in terms of the monomer) essentially the same as reported by the Dutzler group. Similarly, Li et al. find dimers that associate through a large extracellular domain. The key finding here is that under very low calcium, the authors find a dimer between TTYH monomers in which each monomer is in a separate membrane. They offer a reasonable biochemical explanation for why the Dutzler group did not see this (invoking the well known 'stray' calcium from many sources).

The main value of this manuscript is the N=2 finding (i.e. both groups get the same structures, note the same extracellular hydrophobic cavity that may bind some yet to be discovered hydrophobic ligand, and importantly both groups provide evidence that these proteins are not chloride channels as previously proposed). Given this negation of a leading idea, it is important that a second study that independently arrived at the same conclusions be published. The authors however, should take the opportunity to show structural comparisons with the Dutzler structures (report RMSDs, etc.). While there are no surprises expected, such a comparison would add value.

We thank the reviewer for the suggestion and have included new panels to Figure S5 showing overlays between mouse and human TTYH2 and TTYH3 cis dimers with pairwise r.m.s.d.s (1.2 Å in both cases) reported in the legend and text.

The key new finding here is the trans-membrane dimer. The authors argue that one possible reason that this was not seen by Dutzler's group (apart from the stray calcium argument) is that there are differences in the interface between the mouse and human proteins that could explain why one makes these dimers and the other does not. Given availability of the structures of the human TTYHs, the authors are in a position to answer this question directly rather than speculate. They should check if there are sufficient differences in the sequences that make these surfaces that might support this idea. If so, such difference should be noted, and if not, that should also be noted as it would discount the idea that the sequence differences are the cause.

We have included additional text indicating we do not expect sequence differences at the interface account for the difference in observed oligomeric state. 18/19 residues involved are strictly conserved between human and mouse and the one difference (mouse G237 / human A327) is not expected to impact the trans-dimerization interface.

The key point about the trans-disc dimers that is not addressed is that such a low calcium environment is not common in extracellular spaces (or even in 'pre-extracellular spaces such as the ER lumen, Golgi lumen, etc. that serve as calcium stores). Hence, one wonders whether there are any conditions, or places, where TTYH might see this sort of low calcium environment. This point needs to be addressed as these interactions could be equivalent to fortuitous interactions that are made by proteins in protein crystals (i.e. surfaces that can come together under special conditions but that are not physiologically relevant). As the main novelty of this manuscript rests on this unexpected dimer (and it is so important that it is in the title), this point should be addressed clearly.

We agree this is an important point for considering the physiological relevance of monomers and trans-dimers. We do not currently know the Ca^{2+} -dependence of the oligomeric state changes (only that we see differences between samples in 1 mM Ca^{2+} and Ca^{2+} -free conditions), so unfortunately we cannot yet be certain whether TTYH-expressing cells experience sufficiently low $[\text{Ca}^{2+}]_{\text{ext}}$ to trigger such changes in oligomeric state. We have added text to make clear that there are several important unknown factors that will be critical for understanding the physiological context of the oligomeric state changes. For dissociation of cis-dimers into

monomers, these include: (1) the affinity of TTYHs for Ca^{2+} , (2) the related dependance of TTYH oligomeric state changes on $[\text{Ca}^{2+}]_{\text{ext}}$, and (3) the range of $[\text{Ca}^{2+}]_{\text{ext}}$ experienced by TTYH-expressing cells under different conditions. For association of monomers into trans-dimers, these additionally include: (4) the binding constant for trans-dimerization and (5) the surface density of natively expressed TTYHs.